# Spatial Heterogeneity Effects on Meta-Community Stability of Annual Plants from a Coastal Dune Ecosystem

**DOI:** 10.3390/plants12112151

**Published:** 2023-05-29

**Authors:** Pua Bar (Kutiel), Ofir Katz, Michael Dorman

**Affiliations:** 1Department of Geography and Environmental Development, Ben-Gurion University of the Negev, Be’er-Sheva 84105, Israel; kutiel@bgu.ac.il; 2Dead Sea and Arava Science Center, Mt. Masada, Tamar Regional Council, Tamar 86910, Israel; katz.phyt@gmail.com; 3Eilat Campus, Ben-Gurion University of the Negev, Hatmarim Blv, Eilat 8855630, Israel

**Keywords:** spatial scales, mobile dunes, semi-fixed dunes, fixed dunes, open and shrub patches, asynchrony, Israel

## Abstract

Spatial heterogeneity affects plant community composition and diversity. It is particularly noticeable in annual plant communities, which vary in space and time over short distances and periods, forming meta-communities at the regional scale. This study was conducted at the coastal dune ecosystem in Nizzanim nature reserve, Israel. This study aimed to analyze the effect of the spatial heterogeneity, which is expressed in differences in the fixation levels of the dunes and patches outside and beneath the dominant *Artemisia monosperma* shrubs, on the characteristics of the annual plant meta-community and its temporal stability, considering the mechanisms that may affect it. Thirteen dunes were studied: three mobile, seven semi-fixed, and three fixed dunes. Data on the annual plants were collected during the spring seasons of 2006, 2007, 2009, 2014, 2015, and 2016. For each dune, 72 quadrats of 40×40 cm were sampled yearly, with 24 quadrats per slope aspect (windward, leeward, and crest), 12 under the shrub, and 12 in the open. The results indicate that the transition from mobile dunes through semi-fixed to fixed dunes is characterized by an increase in annual plant cover, species richness, species diversity, changes in plant communities, and stability driven by the asynchrony of species population fluctuations. Asynchrony affected the stability of the meta-community of this ecosystem in patches beneath the shrubs but not in the open patches.

## 1. Introduction

A crucial question in the ecology of communities and ecosystems is how they maintain their species composition over time, i.e., how they remain stable and what mechanisms are involved. This question is theoretically and practically relevant, especially in an era of significant environmental changes caused directly and indirectly by humankind [1].

Theory predicts that an ecosystem with higher species richness or species evenness should have greater community stability due to its ability to facilitate greater functional diversity [2,3,4,5]. Such communities can maintain ecosystem functions under environmental changes, because the more diverse they are, the more likely they are to contain tolerant species that can persist or recover under environmental changes [6,7]. Recent studies have shown that asynchrony (the degree to which different units respond dissimilarly through time) matters more than species richness or species evenness, because asynchrony facilitates compensatory dynamics in which different species dominate abundance or productivity at different time periods [8,9]. However, environmental drivers may alter the intricate relationship among richness, synchrony, and stability [9,10,11,12].

Annual plant species, which compose about 50% of the flora in the Eastern Mediterranean Basin, significantly contribute to species diversity in this semi-arid region [13,14,15,16]. They adapt to multiple spatial and temporal changes and respond rapidly to them, so their community composition changes in space and time over short distances and periods. One major biotic factor affecting annual plants and annual plant community composition is facilitation by woody plants acting as ecosystem engineers or nurse shrubs. Woody plants facilitate conditions for smaller plants growing beneath or at the edges of their canopies in various manners. They provide physical protection from wind, sun irradiance and extreme temperatures. In a comparable manner, unpalatable (e.g., thorny or toxic) shrubs deter herbivores not only from themselves but also from plants growing beneath them. Woody plants roots also facilitate edaphic conditions, for example by forming fissures in the soil through which smaller plants can germinate or develop their own root system and through which water can infiltrate the soil. Their root exudates and root detritus can secondarily benefit smaller plants by improving soil structure, organic matter and nutrient availability. Finally, woody plants above-ground litter accumulation can provide more physical protection and nutrition for smaller plants. However, it should be kept in mind that woody species may also be allelopathic and harmful to plants growing beneath them, and that a single woody species can have both positive and negative effects, depending on the mechanism involved. Moreover, differences in shrubs size and canopy compactness, or differences in rainfall amounts from region to region, can have a different effect on the degree of positive facilitation of shrubs on annual plants [17,18,19,20]. Thus, annual plant species diversity and their spatial distribution depend on their relationships with perennial plants and the spatial and temporal heterogeneity of the ecosystem.

Working with various temporal and spatial scales enables us to better understand how information is transferred from finer scales to broader scales (from micro-habitat to macro-habitat) and vice versa, how spatial heterogeneity maintains stability, and what the drivers of stability across spatial and temporal scales are [21,22,23,24,25,26]. Coastal dune systems are a good study case of such scale differences, owing to a spatial heterogeneity among dune fixation states, slope aspect (windward, leeward, crest) and inter-dunal depression, and shrub–open patch types. A study conducted by Bar (Kutiel) and Dorman at Nizzanim Long Term Ecological Research (LTER) coastal nature reserve [27] showed that combining the dune, aspect, and patch scales provides a more complete picture of the processes underlying the changes in annual vegetation during dune fixation, since different processes take place at different scales and stages.

Nizzanim LTER nature reserve, such as most coastal dunes in Israel, offers a good case study for such dynamics. The dunes differ in their fixation state, vegetation cover, and plant and animal species composition [10]. As part of the long-term research scheme in Nizzanim, the dunes are categorized into three types: mobile dunes, semi-fixed dunes, and fixed dunes. The classification of dune mobility state levels is based on the Dune Assemblage Index (DAI), which indicates the affinity of annual plant assemblages to dune fixation [28], and by visual indicators such as dune geomorphic structure and soil characteristics [16,29,30]. Perennial plant cover and their spatial distributions were also considered as indicators for the definition of dune fixation state, based on field measures and aerial photo analyses [28,31].

The mobile dunes represent an arid-like enclave within the Mediterranean district of Israel [31], owing to low water availability and sparse vegetation on these dunes. They are therefore dominated by species adapted to aridity and soil movement, i.e., desert and psammophylous species, such as the Marram grass (*Ammophila arenaria*). In later fixation stages, the soil-fixing shrub *Artemisia monosperma* becomes dominant, replacing the Marram grass alongside other species better adapted to Mediterranean conditions [16,29]. An interesting and important observation is that *A. monosperma* cannot achieve a full (100%) cover on the fixed dunes, since its water demands require maintaining a sparsely vegetated area around it that acts as a water source for the shrub. Therefore, a spatial pattern of nurse plant patches (*A. arenaria* and *A. monosperma* facilitating conditions for annual plants growing beneath them) and open spaces between them is maintained throughout the dune system, albeit in different proportions and constituent species.

A major issue in managing Israel’s coastal dunes, including Nizzanim, is deciding what dune fixation types are desired [32]. Although there is a general consensus that it is of paramount importance to conserve the rare and unique mobile dune habitats, whose areas are decreasing, some people suggest that it is also valuable to maintain the semi-fixed and fixed dunes, because the new types of habitats with the unique features they provide increase regional-scale habitat and species diversity [33,34,35].

We, therefore, find that Nizzanim represents not only a suitable setting for the current study, but is also an appropriate system to demonstrate the potential broader and applicable outcomes of such a study. Understanding the structure of the local annual plants meta-community (a set of interacting communities that are linked by the dispersal of multiple species) and how spatial heterogeneity at both scales, dune and patch types, contributes to its long-term stability, is essential for a better management of this dune system, both in terms of devising conservation priorities and their practical implementation in the field.

The current study aimed to find: (a) How each spatial heterogeneity scale affects the features (cover, species richness, species diversity, and community composition) of the annual plants meta-community in the Nizzanim coastal dune nature reserve? and (b) Is this meta-community stable in time, and if so, what mechanisms drive this temporal stability?

## 2. Results

### 2.1. Cover and Diversity

The average annual plant cover per sampling unit increased sharply and significantly with the increase of the perennial vegetation cover from about 2%, in average, on the mobile dunes, to about 15% on the fixed dunes (Figure 1a, Table 1). Similar trends were found when considering annual plant cover separately in open and beneath-the-shrub patch types (Figure 1b, Table 1). However, in this case, the increase rate in annual plant cover with dune fixation was sharper and higher in open patches compared to patches beneath the shrubs (Figure 1b, Table 2).

The average number of annual plant species per sampling unit increased significantly with perennial vegetation cover, from ∼20 species on mobile dunes to ∼50 species on fixed dunes (Figure 2a, Table 3). Similar tendencies were observed for Shannon and Simpson species diversity indices (Figure 2a, Table 3). However, the tendencies differ when we zoom in on the patch scale. The average number of species per sampling quadrat (40×40 cm) increased significantly in both shrub and open patches, parallel with the increase in perennial vegetation cover (Figure 2b, Table 3). No significant differences were found between patch types (Table 4). However, at the patch scale, Shannon and Simpson species diversity indices significantly increased with perennial plant cover only in the patches beneath the shrubs. In the open patches, the values of both indices were approximately constant with increased perennial cover (Figure 2b, Table 3).

### 2.2. Plant Assemblages

Based on permutation tests, species composition significantly differed among the three dune types (Figure 3a, Table 5) and between the two patch types (Figure 3b, Table 5). In mobile dunes, no difference in annual plant composition was detected between patch types (Figure 4a, Table 5). In contrast, in the two other types of dunes (semi-fixed and fixed), the composition significantly differed among the patch types (Figure 4b,c, Table 5).

### 2.3. Temporal Stability

The temporal stability of annual plant assemblages at the dune scale was neither affected by the cover of perennial plants nor by the species richness and diversity. The only variable that significantly affected the stability of species assemblages was their temporal asynchrony (Figure 5a, Table 6). The stability of annual plant assemblages was only significantly affected by the plant species fluctuation (asynchrony) in beneath-the-shrub patches (Figure 5b, Table 6). However, no significant differences between patch types, in terms of their effects on stability, were detected (Table 7).

## 3. Discussion

Ecosystem stability provides information about the predictability and consistency of ecosystem functioning through time. We used data from a coastal dune system to find predictors of stability at two spatial scales: dune types and patch types. Dunes represent different fixation levels as part of the dune fixation process. Each of the dune types has its characteristic plant community. Simultaneously, the patches beneath and outside the shrubs change and develop distinctive plant communities during the fixation process. The spatial puzzle of these annual plant communities defines the meta-community of this ecosystem. Stability increased due to asynchrony among species populations when moving along the dune and patch type communities.

### 3.1. Cover and Species Diversity

During the dune fixation process, the annual plant cover increases rapidly. The increase is mainly due to the open patches between the shrubs, where the annual plant cover is much higher than under the shrubs. Low annual plant cover, and its moderate increase beneath the shrubs, together with differences in cover among patches within each dune type, were relatively small compared to those in the open patches. These results indicate that the spatial heterogeneity in the open patches between shrubs, in terms of annual plant cover, is higher [16,36] than beneath the shrubs (Figure 1), and that the shrubs do not facilitate annual plant cover beneath them during the dune fixation process [36]. Experimental removal of shrubs conducted at different sites along the coastal dunes in Israel and Australia indicate an increase in annual plant cover, suggesting that shrub canopies inhibit the cover of annual plants in these ecosystems [36,37,38].

All species diversity indices increased significantly with dune fixation. However, when we refer to the patch scales, open vs. beneath the shrubs, we detect differences in the trends of annual plant diversity when comparing the two patch types. Despite the non-significant differences in the number of species between the two patch types, the values of Shannon and Simpson indices were significantly higher for the annual plants beneath the shrubs than in the open. It is particularly evident in the Simpson Index, which over-represents the dominant species. Shannon and Simpson’s indices remain stable in the open patches during the fixation process, while they increase significantly beneath the shrubs. These might result from significant changes in abiotic conditions beneath the shrub during the fixation process [39]. The humus amount, for example, increases significantly beneath the shrubs during the dune fixation process by 2, 4, and 5 times, respectively, compared to the open patches [39], and hence soil maturation proceeds rapidly. Simultaneously, shrub size and structure change significantly [31]. These shrub changes can influence the light fluxes and the number and type of seeds arriving and staying beneath the shrubs [40,41].

Each dune type (mobile, semi-fixed, and fixed) has its own characteristic annual plant assemblage. Moreover, assemblages within each dune type differ between the open and beneath-the-shrub patches (except in mobile dunes), thus illustrating different ecological amplitudes of annual plants in this meta-community driven partly by the heterogeneous physical conditions in this sand ecosystem [17,36,42].

### 3.2. Stability

Annual plant stability at the dune scale is determined directly by the asynchronous fluctuation of species increasing during dune fixation. Zooming in on the patch scale reveals that this increase is mainly due to the higher stability beneath the shrub patch derived from species asynchrony. The number of species beneath the shrubs increases similarly to that in the open patches during the fixation process—an increase in perennial shrub cover and changes in the spatial patterns [28]. However, species diversity remains almost constant in the open patches while it rises significantly beneath the shrubs. The changes in the biotic (and abiotic) conditions beneath the shrubs are higher than at the open patches. This is also apparent in the asynchronous fluctuations between the species and assemblages.

Asynchrony can reflect either heterogeneity in species (functional) responses to environmental conditions (response diversity) or their demographic stochasticity [26,43,44,45]. The degree to which asynchrony links to diversity was also found to be influenced by the environmental condition in both empirical [26,46] and modeled systems [45].

### 3.3. Synthesis

Combining the dune and patch scales and, above all, the regional scale, provides a more complex picture of underlying temporal changes in annual plant communities during dune fixation driven by changes in spatial heterogeneity. The transition from mobile, through semi-fixed, to fixed dunes is characterized by an increase in annual plant cover, species richness and species diversity, changes in plant communities, and stability driven by the asynchrony of species fluctuations. Patches beneath the shrubs are essential for determining the stability of the meta-community at the regional scale. In the coastal dune ecosystem that we studied, shrubs have a significant role responsible for dune fixation, spatial heterogeneity, and the temporal stability of annual vegetation communities. Asynchronous responses among local annual plant communities drive temporal stability linked with species’ populations fluctuating asynchronously across space, stemming from physical and/or competitive differences among local communities.

## 4. Materials and Methods

### 4.1. Study Site

The study area is a coastal dune system located in the Nizzanim LTER site in the southern part of the Israeli Mediterranean coastal plain (31°42${}^{\prime }$–31°44${}^{\prime }$ N, 34°35${}^{\prime }$–34°36${}^{\prime }$ E), covering an area of 20 km${}^{2}$ (Figure 6a,b). The area consists of mobile, semi-fixed and fixed dunes (Figure 7) separated by densely vegetated inter-dune depressions [31].

The climate is Mediterranean, with an annual average temperature of 20 °C and yearly rainfall of 400–500 mm (Figure 6c), falling mainly during winter (November–April). The typical wind direction is southwest, with low wind power expressed by a low drift potential index (147) [47]. Dune heights rarely exceed 20 m above mean sea level.

### 4.2. Vegetation Sampling

Thirteen dunes were studied: three mobile dunes, seven semi-fixed dunes, and three fixed dunes. Since the semi-fixed dunes comprise about two-thirds of the total study area, and variation among them is greater than among mobile and stable dunes, the sampling effort was uneven.

The perennial cover was measured in the field by dividing each dune into three sections: wind-facing slope, crest (top), and slipface. Perennial plant cover for each slope was measured using three 50-m transects running parallel to the length of each slope. For each transect, the aerial cover of all shrubs directly under the transect line was recorded and later pooled per slope. The total cover per slope was then standardized by slope width to give the entire cover per dune. For each dune, we then calculated the average perennial plant cover across all year samples to provide a single measure of the average perennial plant cover for each dune over the entire study period [48]. Perennial plant distribution was measured from aerial photos as described by Rubinstein et al. [28] and Bar (Kutiel) et al. [31].

The mobile parabolic dunes have 5–15% perennial vegetation cover, mainly distributed on the dune crest and the leeward. European beachgrass (*Ammophila arenaria*) is characteristic only of the mobile dunes and is the dominant perennial species along with the wormwood shrub (*Artemesia monosperma*). Semi-fixed dunes have 16–30% perennial plant cover, and fixed dunes have 31–50% perennial plant cover, distributed almost evenly across all slopes. The semi-fixed and the fixed dunes are dominated by wormwood, followed by the desert broom shrub (*Retama raetam*) and perennial herbaceous species.

We collected data on the annual plants during spring (March-April) in the years 2006, 2007, 2009, 2014, 2015, and 2016. For each dune, 72 quadrats of 40×40 cm were sampled yearly, with 24 quadrats per slope aspect (windward, leeward, and crest), 12 under the shrub and 12 in the open, adjacent to the observed shrubs. Quadrats were placed randomly during fieldwork (but alternated between shrub and open patches) within a 100 m${}^{2}$ area at the middle of the wind and the leeward slopes, respectively, and on the dune crest. The quadrats beneath the shrubs were placed only under *Artemisia monosperma*, a key species of the coastal dunes in the Levant [49]. Annual plants were identified to species level, and their relative percentage cover was recorded at each quadrat. See Appendix A for the list of recorded species and their average cover.

### 4.3. Data Analyses

We analyzed annual plant community composition (expressed as % cover) of a total of 80 species recorded. The sites are divided among 13 dunes of 3 dune types (M = mobile, SF = semi-fixed, and F = fixed), 2 patch types (O = open, and B = shrub), and 6 years (2006, 2007, 2009, 2014, 2015, and 2016). We analyzed the data on either the patch scale, where B and O sites were treated separately, or dune scale, where B + O were averaged per dune per year.

Hypotheses regarding effects on the dependent variables, namely annual cover (Figure 1), richness, Shannon index, and Simpson index (Figure 2), and stability (Figure 5), were tested using:Linear regression—when the dependent variable was continuous: namely annual cover (Figure 1, Table 1 and Table 2), Shannon index (Figure 2, Table 3 and Table 4), and community stability (Figure 5, Table 6 and Table 7);Poisson Generalized Linear Model (GLM), when the dependent variable reflects count: namely species richness (Figure 2, Table 3 and Table 4);Beta regression, when the dependent variable was bounded between 0 and 1, namely: Simpson index (Figure 2, Table 3 and Table 4);Permutation tests for Canonical Correspondence Analysis (CCA), when the dependent variable was multivariate community composition (Figure 3 and Figure 4, Table 5).

The independent variables in the regression and GLM models were perennial cover (Figure 1 and Figure 2, Table 1, Table 2, Table 3 and Table 4), or richness, perennial cover, Shannon index, Simpson index, or asynchrony (Figure 5, Table 6 and Table 7). In each case, we fitted one model to the averaged data from B + O patches (Figure 1a, Figure 2a, and Figure 5a), and two more models to the data from B and O patches separately (Figure 1b, Figure 2b, and Figure 5b), see the “group” column in Table 1, Table 3 and Table 6. Furthermore, to evaluate whether the independent variable effect differs among patch type, we fitted models with an interaction term to the combined data from both B and O patches (Table 2, Table 4 and Table 7). We visualized community composition variation among sampling sites using CCA, constrained by dune and patch types. CCA permutation tests were done to evaluate pairwise differences between dune types and patch types in the combined B and O data (Figure 3), and then to evaluate differences between B and O patches in each dune type, separately (Figure 4), see Table 5.

Stability was calculated as the inverse of the coefficient of variation (SD/mean) of total plant cover of the community in dunes and patches over different years [9]. Synchrony was estimated using the “log var ratio” formula, taking the logarithm of the variance (among years) of total community cover divided by the sum of the variance of individual species (see Equation (2) in Ref.  [9]). Positive synchrony values indicate a common response of the species, values close to zero indicate predominantly random fluctuations, while negative values indicate negative covariation between species [9]. In the analyses, we reversed the sign of the “log var ratio” estimates (multiplying by −1), thus referring to asynchrony. Community structure was visualized among dune and patch types using CCA. The significance of the latter constraining environmental effects was evaluated using permutation tests.

### 4.4. Software

The analyses were done in R [50]. Shannon and Simpson diversity indices, as well as ordination analyses and the associated permutation tests, were calculated using package vegan [51]. Beta regression models were fitted using package betareg [52]. Figures were prepared using package ggplot2 [53].

## 5. Conclusions

In this study, we followed the changes in the annual plant during dune fixation on two distinct spatial scales—dunes and patches within dunes. At the dune scale, we identified the general trend of increasing annual plant cover, species richness and diversity, change in plant communities, and their stability driven by the asynchrony fluctuation of species during dune fixation. Analysis of the patch scale revealed that the open patches and the patches beneath the shrub differ. Each one has its characteristic plant community composition and changes in species diversity and stability, which increase sharply beneath the shrubs, as compared to the open patches. The spatial heterogeneity at the nature reserve area, expressed in different dune types and patches, creates a stable meta-community of annual plants, which should be implemented in its management.

## Figures and Tables

**Figure 1 plants-12-02151-f001:**
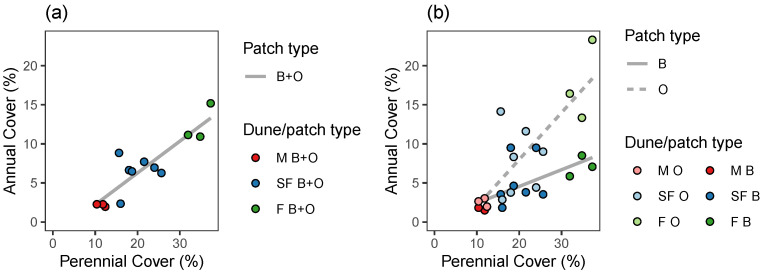
Annual plant cover as a function of perennial plant cover, for each dune (**a**) and for each dune and patch type (**b**). Lines illustrate linear regression fit.

**Figure 2 plants-12-02151-f002:**
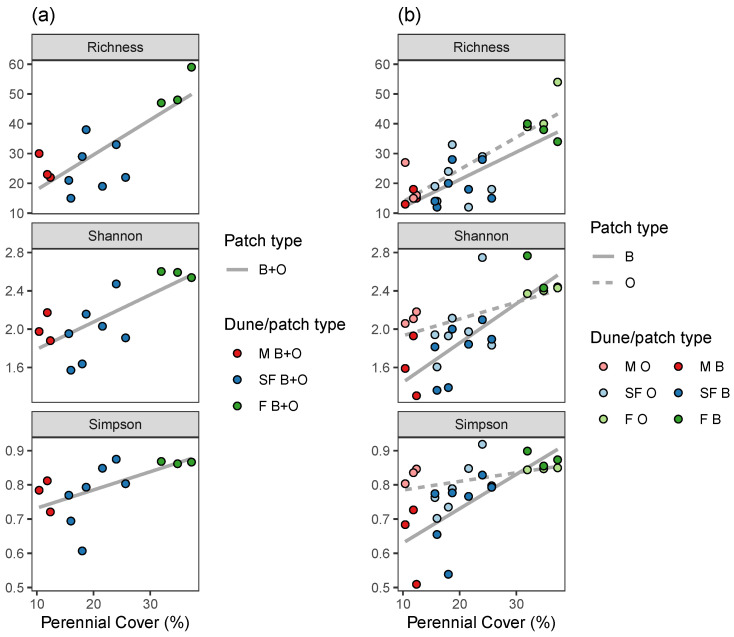
Species diversity metrics (richness, Shannon index, and Simpson index) as a function of perennial cover, for each dune (**a**) and for each dune and patch type (**b**). The lines illustrate the linear regression fit.

**Figure 3 plants-12-02151-f003:**
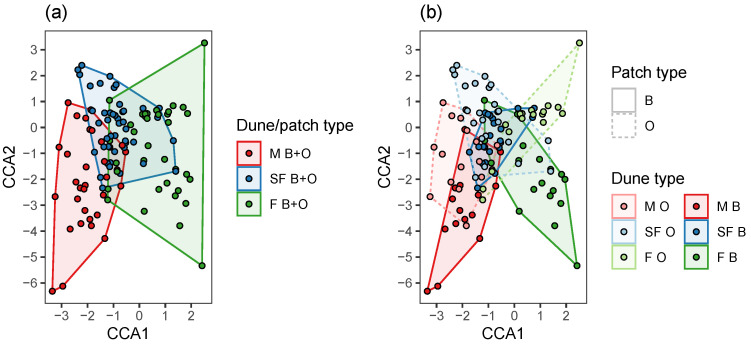
Site scores on axes 1 and 2, using Canonical Correspondence Analysis (CCA) constrained by dune type and patch type for all sites combined. Dune types (**a**) or dune type and patch type combinations (**b**) are marked using convex hull polygons.

**Figure 4 plants-12-02151-f004:**
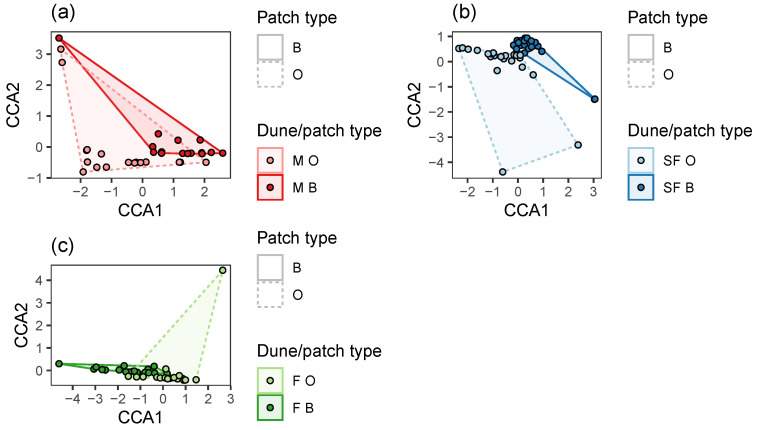
Site scores on axes 1 and 2, using Canonical Correspondence Analysis (CCA) constrained by patch type, separately for Mobile (M) (**a**), Semi-Fixed (SF) (**b**), and Fixed (F) (**c**) dunes. Patch types are marked using convex hull polygons.

**Figure 5 plants-12-02151-f005:**
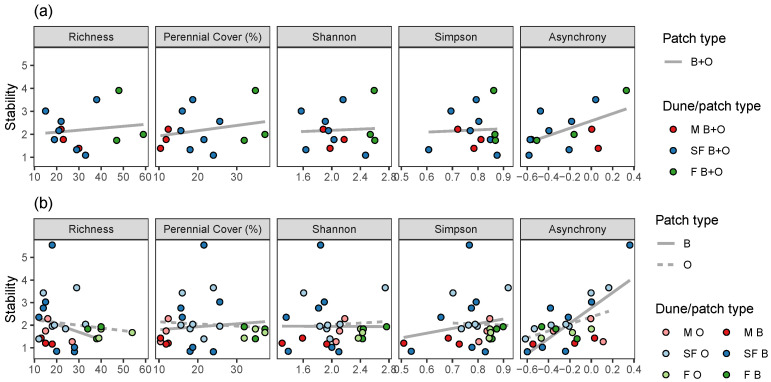
Temporal stability of annual plant assemblages as a function of richness, perennial cover, Shannon and Simpson diversity indices, and asynchrony, for each dune (**a**), and for each dune and patch type (**b**). Lines illustrate linear regression fit.

**Figure 6 plants-12-02151-f006:**
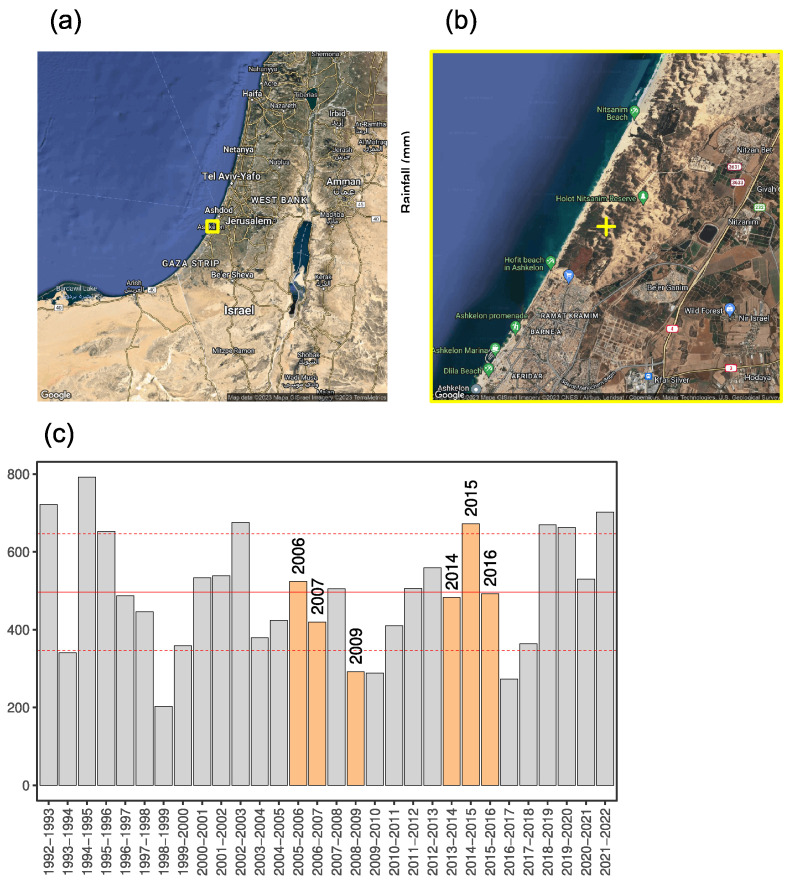
Location (**a**,**b**) and climatic conditions (**c**) of the study area. Location of the Nizzanim nature reserve (**a**), and zoomed-in map where the study site is shown as yellow “+” symbol (**b**). Annual rainfall during the period 1992–2022 in the study site (**c**), where the sampling years 2006, 2007, 2009, 2014, 2015, and 2016 are marked, and horizontal lines mark the average rainfall amount ± one standard deviation.

**Figure 7 plants-12-02151-f007:**
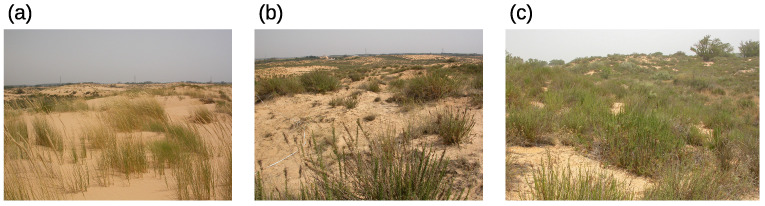
Dune types: (**a**) Mobile (M), (**b**) Semi-Fixed (SF), (**c**) Fixed (F). Photographs by Prof. Pua Bar, 2018.

**Table 1 plants-12-02151-t001:** Effects of perennial cover on annual cover, in both patch types combined (B + O) (Figure 1a), in shrub patches (B), and in open patches (O) (Figure 1b).

Group	Term	Estimate	Std. Error	Statistic	*p*-Value	R${}^{2}$
B + O	(Intercept)	−1.8477	1.4362	−1.2865	0.2247	—
—	Perennial Cover	0.4062	0.0624	6.5135	<0.001	0.7754
B	(Intercept)	0.2796	1.7809	0.1570	0.8781	—
—	Perennial Cover	0.2136	0.0773	2.7621	0.0185	0.3558
O	(Intercept)	−3.9992	3.0680	−1.3035	0.2190	—
—	Perennial Cover	0.5996	0.1332	4.5016	0.0009	0.6162

**Table 2 plants-12-02151-t002:** Effects of perennial cover, patch type, and their interaction, on annual cover (Figure 1b).

Term	Estimate	Std. Error	Statistic	*p*-Value	R${}^{2}$
(Intercept)	0.2796	2.5084	0.1115	0.9123	—
Perennial Cover	0.2136	0.1089	1.9610	0.0627	—
Patch O	−4.2788	3.5474	−1.2062	0.2406	—
Perennial Cover × Patch O	0.3861	0.1540	2.5065	0.0201	0.6176

**Table 3 plants-12-02151-t003:** Effects of perennial cover on richness, Shannon index, and Simpson index, in both patch types combined (B + O) (Figure 2a), in shrub patches (B), and in open patches (O) (Figure 2b).

Group	Dependent	Term	Estimate	Std. Error	Statistic	*p*-Value	R${}^{2}$
B + O	Richness	(Intercept)	2.6380	0.1426	18.5050	<0.001	—
—	—	Perennial Cover	0.0353	0.0055	6.3777	<0.001	0.6201
B + O	Shannon	(Intercept)	1.5069	0.1879	8.0205	<0.001	—
—	—	Perennial Cover	0.0284	0.0082	3.4811	0.0051	0.4809
B + O	Simpson	(Intercept)	0.6727	0.2565	2.6231	0.0087	—
—	—	Perennial Cover	0.0320	0.0118	2.7071	0.0068	—
—	—	(phi)	49.6215	19.3532	2.5640	0.0103	0.4278
B	Richness	(Intercept)	2.2372	0.1691	13.2271	<0.001	—
—	—	Perennial Cover	0.0384	0.0065	5.8975	<0.001	0.6852
B	Shannon	(Intercept)	1.0352	0.2020	5.1242	0.0003	—
—	—	Perennial Cover	0.0410	0.0088	4.6744	0.0007	0.6347
B	Simpson	(Intercept)	−0.0678	0.2844	−0.2385	0.8115	—
—	—	Perennial Cover	0.0552	0.0135	4.0874	<0.001	—
—	—	(phi)	36.0814	14.0262	2.5724	0.0101	0.6425
O	Richness	(Intercept)	2.3795	0.1571	15.1445	<0.001	—
—	—	Perennial Cover	0.0387	0.0060	6.3972	<0.001	0.5875
O	Shannon	(Intercept)	1.7579	0.2009	8.7496	<0.001	—
—	—	Perennial Cover	0.0174	0.0087	1.9969	0.0712	0.1993
O	Simpson	(Intercept)	1.1388	0.2510	4.5369	<0.001	—
—	—	Perennial Cover	0.0160	0.0112	1.4202	0.1555	—
—	—	(phi)	56.0042	21.8686	2.5609	0.0104	0.1404

**Table 4 plants-12-02151-t004:** Effects of perennial cover, patch type, and their interaction, on richness, Shannon index, and Simpson index (Figure 2b).

Dependent	Term	Estimate	Std. Error	Statistic	*p*-Value	R${}^{2}$
Richness	(Intercept)	2.2372	0.1691	13.2271	<0.001	—
—	Perennial Cover	0.0384	0.0065	5.8975	<0.001	—
—	Patch O	0.1423	0.2309	0.6165	0.5376	—
—	Perennial Cover × Patch O	0.0003	0.0089	0.0298	0.9763	0.6393
Shannon	(Intercept)	1.0352	0.2015	5.1382	<0.001	—
—	Perennial Cover	0.0410	0.0087	4.6872	0.0001	—
—	Patch O	0.7227	0.2849	2.5367	0.0188	—
—	Perennial Cover × Patch O	−0.0236	0.0124	−1.9062	0.0698	0.5213
Simpson	(Intercept)	−0.0714	0.2596	−0.2752	0.7832	—
—	Perennial Cover	0.0558	0.0124	4.5153	<0.001	—
—	Patch O	1.2038	0.3827	3.1450	0.0017	—
—	Perennial Cover × Patch O	−0.0400	0.0176	−2.2696	0.0232	—
—	(phi)	43.8577	12.0847	3.6292	0.0003	0.558

**Table 5 plants-12-02151-t005:** Permutation tests for difference in composition between dune types (Figure 3a) and between patch types (Figure 3b) with all dune types combined, and between patch types in each dune type separately (Figure 4).

Group	Comparison	Chi-Square	F	*p*-Value
M+SF+F	M−SF	0.1379	3.8888	0.001
M+SF+F	SF−F	0.1723	4.8585	0.001
M+SF+F	M−F	0.1768	4.9849	0.001
M+SF+F	B−O	0.0981	2.7655	0.001
M	B−O	0.1523	1.5191	0.080
SF	B−O	0.1446	3.0417	0.001
F	B−O	0.1861	2.0053	0.001

**Table 6 plants-12-02151-t006:** Effects of richness, perennial cover, Shannon index, Simpson index, and asynchrony, on stability in both patch types combined (B + O) (Figure 5a), in shrub patches (B), and in open patches (O) (Figure 5b).

Group	Independent	Term	Estimate	Std. Error	Statistic	*p*-Value	R${}^{2}$
B + O	Richness	(Intercept)	1.9257	0.6464	2.9791	0.0125	—
—	—	value	0.0085	0.0192	0.4429	0.6664	0.0718
B + O	Perennial Cover	(Intercept)	1.7048	0.6474	2.6334	0.0233	—
—	—	value	0.0227	0.0281	0.8079	0.4363	0.0298
B + O	Shannon	(Intercept)	1.9158	1.5768	1.2150	0.2498	—
—	—	value	0.1300	0.7366	0.1765	0.8631	0.0878
B + O	Simpson	(Intercept)	1.8108	2.5607	0.7071	0.4942	—
—	—	value	0.4793	3.2156	0.1491	0.8842	0.0887
B + O	Asynchrony	(Intercept)	2.5842	0.2720	9.5009	<0.001	—
—	—	value	1.5928	0.7225	2.2045	0.0497	0.2434
B	Richness	(Intercept)	2.7195	0.9319	2.9183	0.0140	—
—	—	value	−0.0342	0.0381	−0.8974	0.3887	0.0165
B	Perennial Cover	(Intercept)	1.6710	1.0091	1.6560	0.1259	—
—	—	value	0.0130	0.0438	0.2963	0.7725	0.0823
B	Shannon	(Intercept)	1.9796	1.7148	1.1544	0.2728	—
—	—	value	−0.0161	0.8750	−0.0184	0.9856	0.0909
B	Simpson	(Intercept)	0.3912	2.3940	0.1634	0.8732	—
—	—	value	2.0919	3.1771	0.6584	0.5238	0.0495
B	Asynchrony	(Intercept)	2.7659	0.3651	7.5748	<0.001	—
—	—	value	3.3932	1.0361	3.2751	0.0074	0.4477
O	Richness	(Intercept)	2.3561	0.4919	4.7895	0.0006	—
—	—	value	−0.0120	0.0171	−0.7010	0.4979	0.0443
O	Perennial Cover	(Intercept)	2.2448	0.5648	3.9743	0.0022	—
—	—	value	−0.0094	0.0245	−0.3848	0.7077	0.0764
O	Shannon	(Intercept)	1.6396	1.5670	1.0463	0.3179	—
—	—	value	0.1893	0.7288	0.2597	0.7999	0.0843
O	Simpson	(Intercept)	2.4123	3.1214	0.7728	0.4559	—
—	—	value	−0.4540	3.8272	−0.1186	0.9077	0.0895
O	Asynchrony	(Intercept)	2.3645	0.2340	10.1061	<0.001	—
—	—	value	1.5490	0.7344	2.1093	0.0586	0.2233

**Table 7 plants-12-02151-t007:** Effects of richness, perennial cover, Shannon index, Simpson index, and asynchrony, patch type, and their interaction, on stability (Figure 5b).

Independent	Term	Estimate	Std. Error	Statistic	*p*-Value	R${}^{2}$
Richness	(Intercept)	2.7195	0.7581	3.5872	0.0016	—
—	value	−0.0342	0.0310	−1.1031	0.2819	—
—	Patch O	−0.3634	1.0342	−0.3514	0.7286	—
—	value × Patch O	0.0222	0.0395	0.5630	0.5791	0.0635
Perennial Cover	(Intercept)	1.6710	0.8177	2.0435	0.0532	—
—	value	0.0130	0.0355	0.3656	0.7181	—
—	Patch O	0.5738	1.1564	0.4962	0.6247	—
—	value × Patch O	−0.0224	0.0502	−0.4465	0.6596	0.1235
Shannon	(Intercept)	1.9796	1.3895	1.4247	0.1683	—
—	value	−0.0161	0.7090	−0.0227	0.9821	—
—	Patch O	−0.3400	2.6607	−0.1278	0.8995	—
—	value × Patch O	0.2054	1.2714	0.1615	0.8731	0.1322
Simpson	(Intercept)	0.3912	1.9501	0.2006	0.8429	—
—	value	2.0919	2.5879	0.8083	0.4276	—
—	Patch O	2.0211	4.8550	0.4163	0.6812	—
—	value × Patch O	−2.5458	6.0346	−0.4219	0.6772	0.1009
Asynchrony	(Intercept)	2.7659	0.3102	8.9176	<0.001	—
—	value	3.3932	0.8800	3.8557	0.0009	—
—	Patch O	−0.4014	0.4305	−0.9323	0.3613	—
—	value × Patch O	−1.8441	1.2856	−1.4344	0.1655	0.3700

## Data Availability

The data presented in this study are available on request from the corresponding author. The data are not publicly available due to being part of a large database used by the project for various research purposes.

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
