# Peer review of "Spatial Heterogeneity Effects on Meta-Community Stability of Annual Plants from a Coastal Dune Ecosystem"

_plants, 2023, doi:10.3390/plants12112151_

Round 1

Reviewer 1 Report (Previous Reviewer 1)

I revised the new version of the manuscript by Bar et al., which focuses on plant community dynamics in a dune ecosystem of Israel. The manuscript was strongly improved, but I have some comments that are still requiring the attention of the authors. Besides several issues about the syntaxis they used, I am particularly worried about their negative attitude for better describing the statistical methods they used. In this new version of the manuscript, some parts of the statistical procedures are presented in the results, while their description in the methods is not fully proper. Below, I provide specific comments about these and other issues I detected. Once again, I recommend to the authors requesting assistance of a Native English speaker before resubmitting the manuscript.

Line 20: add “species” before “composition”

Lines 47-48: this sentence (Finally, woody ’plants’ above-ground litter …for smaller plants) is not clear. What do the authors mean in this sentence? Rewrite the sentence and make it clearer.

Lines 37-51: The arguments included in these lines have not references supporting them. Please, provide proper reference(s) at the end of each of these sentences for supporting the phenomena you are describing.  

Line 58: replace “versa; how” with “versa; how”. Note the use of a comma instead of a semicolon. Avoid the use of semicolons, as they do not necessarily contribute to improve the understanding of the sentences.

Line 60: replace “good case study for scale difference” with “good study case of such scale differences”

Lines 62: What does LTER mean? This acronym was nor defined in previous sentences (it is defined below, in the next paragraph). Definitions of acronyms are required before they are used in the text and, please, do not abuse of acronyms just for avoiding writing the full words.

Line 63: replace “dune, slope” with “dune slope”. Otherwise, make this sentence clearer for establishing to what processes are referred here.

Lines 66-67: This sentence is repeating what you said in the last sentence of the previous paragraph, when the findings of Bar Kutiel and Dorman are described. Please, reaccommodate the text for avoiding these redundancies.

Lines 88-89: replace “is deciding on which dune fixation types are desired” with “is deciding what dune fixation types are desired”.

Line 89: replace “While there is generally a consensus” with “Although there is general consensus”.

Line 93: remove “at regional scales”. It is redundant with the earlier enunciates of this sentence.

Lines 94-96: is this sentence necessary? What is the relevance of the studied system for supporting the scientific enunciates of the introduction?

Lines 97-98: replace “multiple, potentially interacting species” with “multiple species”.

Line 99: replace “for better management” with “for a better management”.

Lines 100-101: replace “both in terms of devising the correct management strategy (conservation

priorities) and its practical implementation in the field (interface)” with “both in terms of devising conservation priorities and its practical implementation in the field”. Please, avoid unnecessary, excessive, and redundant wording.

Line 105: replace “which mechanism” with “what mechanisms”.

Caption of figure 1: please, improve the caption of this figure. As currently written, it is too hard understanding what figures a, b and c are showing.

Line 122: replace “dividing Each” with “dividing each”. In other words, do not use a capital letter in “Each”.

Lines 132-138: This paragraph contain information that should be moved to the methods, as in these sentences the authors describe the plant cover and the dominant plans species they found at each dune type.

Line 143: replace “but alternately into a shrub or open patches” with “but alternated between shrub and open patches”.

Line 156-157: the hypotheses should be clearly raised at the introduction of the manuscript. Further, I do not fully understand the link that the authors made between the text and figures 3, 4 and 7. I think that this does not meet the standards of scientific writing and, therefore, the authors must rewrite the hypotheses, stating them at the introduction using the proper language.

Lines 158-166: Thess statistical methods must be redacted in a single or paragraph or several paragraphs, instead of just using bullets and making references to figures. I mentioned this in the revision of the former version of the manuscript, but it seems that the authors refuse to provide clear details about their statistical procedures. Here, they must clearly describe what the response variables are and what the predictive variables are, as well as the aim behind each of these analyses. Once again, authors must be prompted to use a proper scientific language and redaction style for their methods.

Line 167: what is a “separate regression”?

Lines 167-171: This first section of this paragraph is extremely confusing. As mentioned in a previous comment, the aim of each statistical analysis (i.e., what you are looking for to tests) must be clearly explained and, please, DO NOT refer to the tables and figures of the results because this generates more confusion. All this section must be rewritten in a clear and proper scientific language. Otherwise, the readers would be unable to know what you did and why you did so.

Lines 196-197: replace “from about 2% on average in the mobile dunes to about 15% in the fixed dunes” with “from about 2%, in average, on the mobile dunes to about 15% on the fixed dunes”

Line 203: replace “cover: from” with “cover, from”, and replace “in mobile dunes” and “in fixed dunes” with “on mobile dunes” and “on fixed dunes”.

Line 207: to improve clarity, here replace “in both patch types” with “in both shrub and open patches”.

Line 208: to improve clarity, here replace “between both” with “between patch types”

Line 210: at the beginning of this line, replace “increase” with “increased”

Line 211: replace “both indices remain the same regardless of perennial cover” with “the values of both indices were approximately constant with increased perennial cover”.

Lines 213-114: this statement (We visualized community composition …by dune and patch types) must me moved to the methods, as it contributes to explain the aim of these statistical analyses.

Line 215: replace “between all three dune types” with “among the three dune types”. The use of the word “between” is restricted to pairwise comparisons, while in this case you are performing comparisons across three treatments and, therefore, the word “among” must be used.

Line 217: replace “composition differences” with “differences in composition”. Please, avoid the use of confusing noun constructions in the text. I invite you to revise the entire manuscript and using a more proper language when apply.

Lines 216-118: this statement (We further examined …of the three dune types separately) must me moved to the methods, as it contribute to explain the aim of these statistical analyses.

Line 223: as you use the word “nor” in this sentence, the phrase “was not” in this line must be replaced by “was neither”.

Line 224: replace “perennial plants’ coverage” with “the cover of perennial plants”. Avoid the use of confusing noun constructions in the text.

Line 225-126: replace “impacted the species assemblages’ stability was the temporal asynchrony of species fluctuations” with “affected the stability of species assemblages was their temporal asynchrony”.

Lines 226-227: What do you mean with “A similar trend regarding the two types of patches was found”? The syntaxis and grammar of this sentence is strange, and its meaning is not fully clear. Please, rewrite in a clearer manner.

Line 227: replace “annual plant assemblages’ stability” with “the stability of annual plant assemblages”. Perform this kind of changes throughout the entire manuscript and, please, AVOID using confusing noun constructions in the text.

Line 237: replace “their distinct communities” with “distinctive plant communities”.

Line 240: replace “dune and the patch-type communities” with “dune type and patch type communities”.

Line 245: replace “between patches” with “among patches”.

Line 2149: replace “plant coverage” with “plant cover”. Also revise the entire manuscript looking for this mistake, as I detected it in several sentences.

Lines 253-254: remove “expressed by increased perennial plant cover” because it makes no sense in terms of changes in species diversity. Otherwise, please describe better the overall idea you want to expose.  

Line 255: replace “we can see” with “we detected”. In this line, also replace “annual plants diversity” with “annual plant diversity” or “diversity of annual plants”.

Line 257: replace “the increase rates” with “the values”, as there is no rates here.

Line 259: here, the authors indicate that the Simpson index “over-represents the dominant species”, but they do not provide any citation supporting such a statement. I worked with the Simpson and Shannon diversity indices before and realized that both of them are highly sensitive to dominance or decreased community evenness, and there are several explaining this. I recommend to the authors to perform a literature revision for citing a previous study dealing with this issue of the indices of proportional diversity.

Line 271: what do you mean with “amplitudes of annual plants”? I cannot imaging how an annual plant can have “amplitude”. Please, make this point clearer rewriting the sentence and using adequate words.

Line 276: replace “due to the stability” with “due to the higher stability” (it is higher stability, am I right?

Line 277: replace “similarly to” with “similarly to that in”.

Line 280: replace “while rising” with “while it rises”.

Line 280-282: the message of the last sentence of this paragraph is not fully clear. Please rewrite and make your point clearer.

Line 283-286: The sentences of this paragraph are out of context. How this relate with your findings? Please, develop the idea a bit more.

Line 295: replace “in determining” with “for determining”.

Line 295-296: replace “In a coastal dune ecosystem, like the one we studied” with “In the coastal dune ecosystem that we studied”. At this stage, I think that you cannot make your findings extensive to all other similar dune ecosystems.

Lines 306-307: replace “open patch” and “patch beneath” with “open patches ” and “patches beneath”, respectively.

Line 309: replace “shrubs compared” with “shrubs, as compared”

See comments to the authors. The language must be improved and I made some suggestions for changes in teh text.

Author Response

I revised the new version of the manuscript by Bar et al., which focuses on plant community dynamics in a dune ecosystem of Israel. The manuscript was strongly improved, but I have some comments that are still requiring the attention of the authors. Besides several issues about the syntaxis they used, I am particularly worried about their negative attitude for better describing the statistical methods they used. In this new version of the manuscript, some parts of the statistical procedures are presented in the results, while their description in the methods is not fully proper. Below, I provide specific comments about these and other issues I detected. Once again, I recommend to the authors requesting assistance of a Native English speaker before resubmitting the manuscript.

Line 20: add “species” before “composition”

Done.

Lines 47-48: this sentence (Finally, woody ’plants’ above-ground litter …for smaller plants) is not clear. What do the authors mean in this sentence? Rewrite the sentence and make it clearer.

Lines 37-51: The arguments included in these lines have not references supporting them. Please, provide proper reference(s) at the end of each of these sentences for supporting the phenomena you are describing.

Line 58: replace “versa; how” with “versa; how”. Note the use of a comma instead of a semicolon. Avoid the use of semicolons, as they do not necessarily contribute to improve the understanding of the sentences.

Done.

Line 60: replace “good case study for scale difference” with “good study case of such scale differences”

Done.

Lines 62: What does LTER mean? This acronym was nor defined in previous sentences (it is defined below, in the next paragraph). Definitions of acronyms are required before they are used in the text and, please, do not abuse of acronyms just for avoiding writing the full words.

Done.

Line 63: replace “dune, slope” with “dune slope”. Otherwise, make this sentence clearer for establishing to what processes are referred here.

Done.

Lines 66-67: This sentence is repeating what you said in the last sentence of the previous paragraph, when the findings of Bar Kutiel and Dorman are described. Please, reaccommodate the text for avoiding these redundancies.

Lines 88-89: replace “is deciding on which dune fixation types are desired” with “is deciding what dune fixation types are desired”.

Done.

Line 89: replace “While there is generally a consensus” with “Although there is general consensus”.

Done.

Line 93: remove “at regional scales”. It is redundant with the earlier enunciates of this sentence.

Done.

Lines 94-96: is this sentence necessary? What is the relevance of the studied system for supporting the scientific enunciates of the introduction?

Lines 97-98: replace “multiple, potentially interacting species” with “multiple species”.

Done.

Line 99: replace “for better management” with “for a better management”.

Done.

Lines 100-101: replace “both in terms of devising the correct management strategy (conservation

priorities) and its practical implementation in the field (interface)” with “both in terms of devising conservation priorities and its practical implementation in the field”. Please, avoid unnecessary, excessive, and redundant wording.

Done.

Line 105: replace “which mechanism” with “what mechanisms”.

Done.

Caption of figure 1: please, improve the caption of this figure. As currently written, it is too hard understanding what figures a, b and c are showing.

Done.

Line 122: replace “dividing Each” with “dividing each”. In other words, do not use a capital letter in “Each”.

Done.

Lines 132-138: This paragraph contain information that should be moved to the methods, as in these sentences the authors describe the plant cover and the dominant plans species they found at each dune type.

This sentence is already in the Methods.

Line 143: replace “but alternately into a shrub or open patches” with “but alternated between shrub and open patches”.

Done.

Line 156-157: the hypotheses should be clearly raised at the introduction of the manuscript. Further, I do not fully understand the link that the authors made between the text and figures 3, 4 and 7. I think that this does not meet the standards of scientific writing and, therefore, the authors must rewrite the hypotheses, stating them at the introduction using the proper language.

Lines 158-166: Thess statistical methods must be redacted in a single or paragraph or several paragraphs, instead of just using bullets and making references to figures. I mentioned this in the revision of the former version of the manuscript, but it seems that the authors refuse to provide clear details about their statistical procedures. Here, they must clearly describe what the response variables are and what the predictive variables are, as well as the aim behind each of these analyses. Once again, authors must be prompted to use a proper scientific language and redaction style for their methods.

We edited the paragraph to make it clearer and specified what the dependent variables are.

Line 167: what is a “separate regression”?

We removed the word "separate".

Lines 167-171: This first section of this paragraph is extremely confusing. As mentioned in a previous comment, the aim of each statistical analysis (i.e., what you are looking for to tests) must be clearly explained and, please, DO NOT refer to the tables and figures of the results because this generates more confusion. All this section must be rewritten in a clear and proper scientific language. Otherwise, the readers would be unable to know what you did and why you did so.

Lines 196-197: replace “from about 2% on average in the mobile dunes to about 15% in the fixed dunes” with “from about 2%, in average, on the mobile dunes to about 15% on the fixed dunes”

Done.

Line 203: replace “cover: from” with “cover, from”, and replace “in mobile dunes” and “in fixed dunes” with “on mobile dunes” and “on fixed dunes”.

Done.

Line 207: to improve clarity, here replace “in both patch types” with “in both shrub and open patches”.

Done.

Line 208: to improve clarity, here replace “between both” with “between patch types”

Done.

Line 210: at the beginning of this line, replace “increase” with “increased”

Done.

Line 211: replace “both indices remain the same regardless of perennial cover” with “the values of both indices were approximately constant with increased perennial cover”.

Done.

Lines 213-114: this statement (We visualized community composition …by dune and patch types) must me moved to the methods, as it contributes to explain the aim of these statistical analyses.

Done.

Line 215: replace “between all three dune types” with “among the three dune types”. The use of the word “between” is restricted to pairwise comparisons, while in this case you are performing comparisons across three treatments and, therefore, the word “among” must be used.

Done.

Line 217: replace “composition differences” with “differences in composition”. Please, avoid the use of confusing noun constructions in the text. I invite you to revise the entire manuscript and using a more proper language when apply.

Done.

Lines 216-118: this statement (We further examined …of the three dune types separately) must me moved to the methods, as it contribute to explain the aim of these statistical analyses.

Done.

Line 223: as you use the word “nor” in this sentence, the phrase “was not” in this line must be replaced by “was neither”.

Done.

Line 224: replace “perennial plants’ coverage” with “the cover of perennial plants”. Avoid the use of confusing noun constructions in the text.

Done.

Line 225-126: replace “impacted the species assemblages’ stability was the temporal asynchrony of species fluctuations” with “affected the stability of species assemblages was their temporal asynchrony”.

Done.

Lines 226-227: What do you mean with “A similar trend regarding the two types of patches was found”? The syntaxis and grammar of this sentence is strange, and its meaning is not fully clear. Please, rewrite in a clearer manner.

Done.

Line 227: replace “annual plant assemblages’ stability” with “the stability of annual plant assemblages”. Perform this kind of changes throughout the entire manuscript and, please, AVOID using confusing noun constructions in the text.

Done.

Line 237: replace “their distinct communities” with “distinctive plant communities”.

Done.

Line 240: replace “dune and the patch-type communities” with “dune type and patch type communities”.

Done.

Line 245: replace “between patches” with “among patches”.

Done.

Line 2149: replace “plant coverage” with “plant cover”. Also revise the entire manuscript looking for this mistake, as I detected it in several sentences.

Done.

Lines 253-254: remove “expressed by increased perennial plant cover” because it makes no sense in terms of changes in species diversity. Otherwise, please describe better the overall idea you want to expose.

Done.

Line 255: replace “we can see” with “we detected”. In this line, also replace “annual plants diversity” with “annual plant diversity” or “diversity of annual plants”.

Done.

Line 257: replace “the increase rates” with “the values”, as there is no rates here.

Done.

Line 259: here, the authors indicate that the Simpson index “over-represents the dominant species”, but they do not provide any citation supporting such a statement. I worked with the Simpson and Shannon diversity indices before and realized that both of them are highly sensitive to dominance or decreased community evenness, and there are several explaining this. I recommend to the authors to perform a literature revision for citing a previous study dealing with this issue of the indices of proportional diversity.

Line 271: what do you mean with “amplitudes of annual plants”? I cannot imaging how an annual plant can have “amplitude”. Please, make this point clearer rewriting the sentence and using adequate words.

Line 276: replace “due to the stability” with “due to the higher stability” (it is higher stability, am I right?

Done.

Line 277: replace “similarly to” with “similarly to that in”.

Done.

Line 280: replace “while rising” with “while it rises”.

Done.

Line 280-282: the message of the last sentence of this paragraph is not fully clear. Please rewrite and make your point clearer.

Line 283-286: The sentences of this paragraph are out of context. How this relate with your findings? Please, develop the idea a bit more.

Line 295: replace “in determining” with “for determining”.

Done.

Line 295-296: replace “In a coastal dune ecosystem, like the one we studied” with “In the coastal dune ecosystem that we studied”. At this stage, I think that you cannot make your findings extensive to all other similar dune ecosystems.

Done.

Lines 306-307: replace “open patch” and “patch beneath” with “open patches ” and “patches beneath”, respectively.

Done.

Line 309: replace “shrubs compared” with “shrubs, as compared”

Reviewer 2 Report (Previous Reviewer 2)

I have revised this manuscript before (ID 2227052 - reviewer 2) and made my comments that have been well answered in the current manuscript by the Authors.

I think the manuscript is ready to publication in the present form after proofreading.

I tried to check typos as well, only found one in line 122, where the word "Each" should be written with a lower case 'e'.

Don't know which package Authors used for Poisson-regression, but citing the appropriate package  (maybe glmmTMB or lmer as I recall) in the text would be good.

Author Response

I have revised this manuscript before (ID 2227052 - reviewer 2) and made my comments that have been well answered in the current manuscript by the Authors.

I think the manuscript is ready to publication in the present form after proofreading.

I tried to check typos as well, only found one in line 122, where the word "Each" should be written with a lower case 'e'.

Done.

Don't know which package Authors used for Poisson-regression, but citing the appropriate package  (maybe glmmTMB or lmer as I recall) in the text would be good.

Poisson regression was fitted using the base function 'glm', so the R program citation already covers it.

This manuscript is a resubmission of an earlier submission. The following is a list of the peer review reports and author responses from that submission.

Round 1

Reviewer 1 Report

The manuscript by Bar et al. focuses on plant community dynamics in a dune ecosystem of Israel. Although the approach used by the authors is not fully novel, some interesting patterns emerge. Nevertheless, the manuscript must be fully rewritten to make it easier to understand, reorganizing the information provided in its sections. In the introduction, for instance, the authors require stating a much more solid conceptual and theoretical framework, as it currently does not clearly define the concepts used in other sections of the manuscript and it does not fully link these concepts with the processes studied. The methods, on the other hand, contain much information that should be moved to the introduction for justifying the selection of the study system. Further, some information is missed in the methods, where some statistical procedures were omitted, while other information suddenly appear in other sections, as occurs on the results with the outputs of analyses that are not described in the methods. For these seasons, while the discussion is too speculative, the validity of the conclusions is hard to evaluate. The manuscript would also benefit with the revision by a native English speaker with experience in the redaction style required for writing scientific articles. On this latter issue, I made some suggestions in my specific comments, but the authors must fully revise the manuscript and avoid the use of confusing wording.

General comment on Title: The title must be shortened and better reflect the aim of the study.

Comment 01: The title is too long. I recommend changing the title to “Spatial heterogeneity effects on meta-community stability of annual plants from a coastal dune ecosystem” or something else.

General comment on the Abstract: The abstract is fine and just a few issues should be addressed by the authors to improve clarity.

Comment 02, Abstract, line 03: add the word “type” after “of the dune”. This makes the sentence clearer (even if the word type is repeated later in “patch types”).

General comment on the introduction: The introduction is too brief, as it only provides general ideas about the processes that the authors want to study. The statements of several paragraphs are poorly redacted. This entire section must be revised and rewritten to state a solid theoretical framework about how plant communities can maintain their composition over time through the different plant-plant interactions, species asynchrony and the other factors that the authors mention across different spatiotemporal scales. This would allow them rising at least a solid hypothesis by the end of the introduction.

Comment 03, Introduction, line 17: replace “time, i.e., remain stable, and by which mechanisms” with “time, i.e., how they remain stable and what mechanisms are involved”.

Comment 04, Introduction, line 19: replace “man” with “humans” for using an inclusive language (the word “man”, in this context, may be interpreted as a genera issue that I recommend avoiding). Perform the same changes in the other sections of the manuscript if necessary.

Comment 05, Introduction, line 44-46: In this sentence, the authors attempt to mention how a compensatory dynamic works in favor of the diversity and stability of plants communities. Nevertheless, the description they provide of these processes is too synthetic, while it deserves a bit extended explanation. I recommend improving this description better establishing how species asynchrony constitutes a compensatory dynamic and, if possible, providing a brief example.

Comment 06, Introduction, line 47-49: here, the authors state that “studies in ecology are sensitive to differences among spatial and temporal scales”. I think that the authors are somewhat confused about this argument. Considering that a scale is the number of units (spatial or temporal) that information requires to take the shape of a message understandable for a receiver, the studied phenomena (but not the study itself) is sensitive to the scale. As currently written this paragraph make no fully sense because of this confusion and, further it is not clear why the authors mention that “some phenomena and mechanisms are observable only at certain scales, whereas others exhibit contrasting patterns at various scales”. The authors must develop the idea they want supporting with these arguments, as it is not clear at all where they are going with this.

Comment 07, Introduction, line 50-59: Here, the authors introduce the study system. Nevertheless, as currently written, the reasons for selecting this study system are not clear. If the authors are interested in study how plant communities maintain their composition over time (as stated in the first paragraph of the introduction), they must better justify the selection of the studied system, also describing how it is useful for advancing in our knowledge about these issues. This would allow to the authors rising a hypothesis that can be tested with the data they gathered. Further, in line 54, the word “metacomunity” is introduced, but the authors never stated in previous sentences of paragraphs what this word means. I guess that it is a system of communities that exchange species following random or directional dispersal processes, but this is not clear. To improve the conceptual framework of the study, the authors must clearly state what they understand by metacommunity and provide adequate references to support these arguments.

Comment 08, Introduction, line 60-63: Put the first sentence of these lines at the end of the previous paragraph and state the objectives of the study sequentially (they are not questions) – e.g.: The current study aimed to determine: (1) what factors affect annual community stability; (2) whether these plant communities stable over time, and if so, what mechanisms drive their temporal stability. Please consider that these objectives should be possibly rewritten if my previous comments are attended.

General comments on the methods: The methods must be strongly improved. Some sections of the methods should be moved to the introduction, as these arguments support the relevance of selecting the dunes for this study (see my comments below). Further, there is confusion about what the authors define as spatial scales and, thus, it is not clear how the data gathered at different spatial scales are treated statistically.

Comment 09, Methods, line 74: replace “and the coastal sand dunes in Israel in general” with “as most coastal sand dunes of Israel”.

Comment 10, Methods, line 74-98: perhaps the contents of these paragraph should be moved to the last paragraph of the introduction, as these arguments well-support the selection of the study system.

Comment 11, Methods, line 81: remove “Hence” at the beginning of this paragraph.

Comment 12, Methods, line 93: remove “found on mobile dunes”, it is unnecessary.

Comment 13, Methods, line 95: replace “that it maintains” with “maintaining”.

Comment 14, Methods, line 96-98: here the authors state that there are nurse plants on the dunes, but they never stated before what are the facilitator and the beneficiary species. In the methods, it would be important to know what of the species mentioned in this paragraphs act as nurses for other species.

Comment 15, Methods, line 99: remove “Thus” at the beginning of this paragraph. Alternatively, merge this paragraph with the previous one. Further, it is not clear what the authors mean with “two scales of spatial heterogeneity”, as I can understand that a landscape composed by more than a single habitat type is heterogeneous, but the space within each habitat type is homogeneous. Thus, the authors have two spatial scales only (the habitat and the landscape), while the word “heterogeneity” is not required here. To improve clarity about this, I recommend to the authors reading the following two articles Wright, Jones & Flecker, 2002, Oecologia 132, 96–101 (https://doi.org/10.1007/s00442-002-0929-1) and Badano, Jones, Cavieres & Wright, 2006, Oikos 15, 369-385 (https://doi.org/10.1111/j.2006.0030-1299.15132.x). Both papers deal with the structure and diversity of communities at the habitat and the landscape scale.

Comment 16, Methods, line 101-202: replace “including in Nizzanim, is deciding on which dune stabilization types are desired” with “including Nizzanim, is deciding what dune stabilization types are desired”.

Comment 17, Methods, line 103: replace “the paramount importance is” with “is of paramount importance”.

Comment 18, Methods, line 104-106: replace “some suggest that there is also value in maintaining the semi-fixed and fixed dunes as new types of habitats with their own unique features that increase regional-scale habitat and species diversity” with “some people suggest that it is also valuable maintaining the semi-fixed and fixed dunes because of the new types of habitats with unique features they provide increase habitat and species diversity at regional scale”.

Comment 19, Methods, line 107: replace “there have” with “there were”.

Comment 20, Methods, line 111: remove “any” before “clear”.

Comment 21, Methods, line 116-118: this first sentence of the paragraphs should be removed because it is too pretentious and it is not fully related with the aims of the study. If the authors want to highlight the relevance and novelty of the study is OK, but they must do that in the introduction, providing a solid conceptual framework founded on scientific arguments.

Comment 22, Methods, line 118-119: here the authors use the word metacommunity again, but it is not clear what they mean with this word. Further, at least for me, it is not clear what “spatial heterogeneity” means. I must insist that the authors must clearly define the concepts with which they are working in the introduction.

Comment 23, Methods, line 126-127: What do the authors mean with “semi-fixed dunes…. vary more than mobile and stable dunes”? What are the features that make semi-fixed dunes more “variable” than mobile and stable dunes? It is not clear, but I think that the explanation is in the next paragraph. If so, merge these two paragraphs in a single one for facilitating the reading and the understanding. Otherwise, provide a more detailed explanation.

Comment 24, Methods, line 141-147: The content of this paragraph seems like a summary of the results. Perhaps it would be convenient moving this information to the results.

Comment 25, Methods, line 149-150: Why is coverage of 80 species reported only? Are these all the species detected in the dunes? Further 13 dunes x 2 habitats x 6 years = 156 sites, why are data from 117 sites reported only?

Comment 26, Methods, line 151-156: this information should be redacted in the form of a paragraph, instead of just providing bullets. The authors must be prompted to use a proper scientific language and redaction style.

Comment 27, Methods, line 157-158: merge the first of these paragraphs with the next one. Avoid using single-sentenced paragraphs. Further, this explanation about how plant community stability was calculated is a bit confusing (because it is too short) and more details must be provided to make clearer this method.

Comment 28, Methods, line 159-162: as mentioned in the previous comment (see comment 26), the methods used to compute plant community variables must be explained better. As currently written, is too hard to understand the metrics that were computed, which data was used to make these computations, and what was the aim behind performing these calculations (i.e., how were they used and for what were they used). All this section requires mor extensive and clear explanations.

Comment 29, Methods, line 136-165: what data was used to make the CCA analysis? This must be clearly specified to know the aim of the analysis, and the analysis must be described better.

Comment 30, Methods, line 166-167: why were the Simpson and Shannon-Weaver indices computed? What was the aim behind computing these indices? What statistical methods were used to compare these data (Simpson and Shannon-Weaver indices) among communities and across spatial scales? These issues are not described at any place of the methods.

General comments on the results: Some results reported by the authors make no sense. This is because they report, for instance, results or regression analyses whose procedures and aim were not described in the methods. For attending these issues, the authors must improve the description of the methods, as well as the description of results.

Comment 31, Results, line 170-177 and figure 2: in the text and the figure, the authors describe the results of regression analyses. However, the aim of these regression analyses is nor described in the methods. Are these simple linear regression analyses (as it seems in Figure 2a) or multiple regression analyses with categorical variables (as it seems in Figure 2b)? If the aims, procedures and data used to perform the statistical analyses are not clearly established, the results are meaningless.

Comment 32, Results, caption of figure 2: please, rewrite the caption of this figure and clearly indicate what are these two panels showing. I am unable to interpret the results contained in this figure with the information provided in this caption.

Comment 33, Results, line 176-186 and Figure 3: in this paragraph, the same queries than in comment 29 apply. It is neither clear how the data used for these analyses were gathered not the procedures used for these analyses (see my comments on the methods), and the aim of the analyses it is not fully clear. All these issues must be better attended in the methods. Further, the values of these species diversity metrics (richness, Shannon-Wiener index and Simpson index) are usually positively correlated among them. Thus, why are reported all these metrics? Is there is some special, and scientifically valid, reason for determining how richness, Shannon-Wiener index and Simpson index relate with perennial plant cover? At least for me, it would be more interesting assessing how the Shannon-Wiener index relates with richness, as this would reveal whether dominant species are downweighting the proportional diversity. I recommend to the authors reading the books of Anne Magurran about these issues (Ecological diversity and its measurement  and Measuring Biological Diversity) and the article of Stirling & Wilsey, 2001, The American Naturalist 158, 186-299 (https://doi.org/10.1086/321317).

Comment 34, Results, line 180-190: The permutation test mentioned here are not described in the methods.

Comment 35, Results, line 198: Why were species richness and diversity included as predictive variables in the analyses of the temporal stability of annual plant assemblages? Why are they important predictive viables? For sustaining this statement, the authors must answer these questions and providing solid arguments in the methods.

General comment on the discussion: The discussion is too speculative and focuses on discussing the results of other studies rather than explaining the results of obtained by the authors. I recommend fully rewriting the discussion focused on the results obtained in the study.

Comment 36, Discussion, line 205: starting the discussion with a subsection named “Overview” is not adequate. Please, change the name of this subsection.

Comment 37, Discussion, line 206-209: This first statement of the discussion is not fully clear, as in the results do not show the multi-scalar approach (dune vs. patch type) that the authors are mentioning. For this, they must describe the methods and results better in order to reach solid conclusions.

Comment 38, Discussion, line 209-219: All these sentences are addressed to explain and justify the first sentence of the discussion. However, they are not addressed to explain the multi-scalar approach (dune vs. patch type) of the authors.

Comment 39, Discussion, line 223-225: here, the authors state that “Hence, annual plants share the dune area with the shrubs even when the dunes are stabilized and are in a state of dynamic equilibrium with the surrounding environment”, but they did not test whether plant communities on the dunes are in a dynamic equilibrium where richness is constant because of species turnovers (which is the definition of dynamic equilibrium according with the Theory of Biogeography of MacArthur and Wilson, 1967). Thus, this concussion of speculative because it is not supported by the results.

Comment 40, Discussion, line 226-228: what is a “fine-grain plant community scales”? The authors must avoid using this kid of confusing terminology. After that, the authors state that “Communities change between dune types and between patches, open and beneath shrubs, forming a meta-community on the regional scale”. This is the first time that a definition of metacommunity is provided, but the explanation provided in this sentence is incomplete and I cannot see how it is supported by the results reported by the authors.

Comment 41, Discussion, line 230: here, the authors state that “During the dune fixation process, the annual plant cover increases rapidly”. However, they did not show any results indicating this patters. Perhaps a simple regression analysis could be conducted for showing how vegetation cover increases over time (the authors have data of consecutive years) in the different dune types.

Comment 42, Discussion, line 234: replace “These indicate” with “These results indicate”.

Comment 43, Discussion, line 234: replace “These indicate” with “These results indicate”.

Comment 44, Discussion, line 235: The statement “the spatial heterogeneity in the open patches between shrubs is higher” is not clear because it is hard to identify because the relationship between plant cover and heterogeneity is not clearly defined at any section of the manuscript. This the main reason by which I recommend to the authors rewriting the manuscript better describing the concepts they use and the process they evaluate.

Comment 45, Discussion, line 237-239: perhaps shrub removal increases the cover of annual plant cover, but: are annuals important for stabilizing dunes or are they involved in key ecosystems processes?

Comment 46, Discussion, line 241-254: All this paragraph is speculative, as it discusses the findings of other authors that are not fully related with the results of the authors. I suggest removing it and focusing the discussion on the findings of the study.

Comment 47, Discussion, line 255-256: here, the authors state that “All species diversity indices increase significantly with dune fixation, expressed by increased perennial plant cover”. Nevertheless, for better supporting this argument, at the introduction the authors should develop the theoretical support behind this positive relationship between diversity-dune fixation-plat cover. Otherwise, this statement is speculative.

Comment 48, Discussion, line 257: what do the authors mean with “we can see differences in annual plants behavior”? How can plants “behave”? Please, avoid this king of confusing wording through the text.

Comment 49, Discussion, line 260-261: here, the authors state that “It is particularly evident in the Simpson Index, which over-represents the dominant species”. Nevertheless, they never indicated why these two diversity indices are used (at least for me, this is redundant) or why the information they provide is relevant. The authors should take into account that the Shannon-Wiener index is a measure of information (e.g., bits), while the Simpson index is a probability. Thus, they are not comparable in terms of the campout of diversity contained in a community. On this issue, the authors must revise their methods and state why these indices were used and how they were compared within and across spatial scales. Otherwise, this section of the discussion makes no sense.

Comment 50, Discussion, line 273-278: what is the aim of this subsection of the discussion (Plant assemblages)? This a short paragraph that just repeat the results, while it does not contribute to understand what these result mean. The authors must better explain these results or removing this section.

Comment 51, Discussion, line 280-281: for making this statement valid, the authors must clearly state in the introduction what species asynchrony is.

Comment 52, Discussion, line 281-285: these sentences (“Zooming in on the patch scale reveals … from species asynchrony” and “As noted, the number of species … the spatial patterns”) are extremely confusing. Please, rewrite it using a clearer language.

Comment 53, Discussion, line 287: what “abiotic measures” did the authors take? This is not explained in the methods and, thus, this concussion is not fully valid.

Comment 54, Discussion, line 293-307: all this section is speculative because the results does not fully support the arguments provided here. I recommend to the authors revising their results and focus the discussion on their findings.

General comment on the conclusions: this section must be rewritten focusing it on the findings of the authors.

Comment 55, Conclusions, line 309-311: here the authors state that state “We have demonstrated how entity-defined multi-scalar analysis can promote ecological understanding by integrating the unique data and knowledge that can be acquired at each scale”. This statement is too ambitious, as the results of the authors are not clear enough for supporting it. This sentence must be rewritten avoiding grandiloquence and confusing terminology that was not clearly defined by the authors in the former sections.

Comment 56, Conclusions, line 312: replace “two distinct scales” with “two distinct spatial scales”. Also replace “On the dune scale” with “At the dune scale.

Comment 57, Conclusions, line 314: once again, I recommend better establishing in the introduction what “asynchrony fluctuation of species” means. Otherwise, this statement makes no sense.

Comment 58, Conclusions, line 315: replace “Analysis of the patch scale” with “At the patch scale”.

Comment 59, Conclusions, line 316: with “Each one has its characteristic plant community” I guess that the authors are referring to the different patches within dunes, but not to differences between the dune scale and the patch scale, am I right? If so, please use a clearer language for explaining this.

Comment 59, Conclusions, line 318-320: to support such a conclusion, the authors must revise my comment on the methods and the results. Further, what does “a prosperous and stable meta-community” means?

Reviewer 2 Report

Dear Authors,

The article is interesting and provides a good and catchy overview about the complexity of processes that shape plant communities in the studied dune ecosystem. The study attempts to depict the simultaneous spatial and temporal patterns, but I fear that more complex statistical methods are needed for a correct evaluation. However, considering the whole study, even the simple linear regression approach can be adequate (though not perfect) for the actual situation since it can clearly describe the main relationships and tendencies that drives community stability and asynchrony.

I recommend to publish this article in the special issue after a minor revision, please find my specific comments below:

1. Figure 1a): The map seems too small and it is hard to read the most important locations, thereby hard to localize the study site. Recommend to "zoom in" a bit or try to generalize the map to only concentrate on the most important details in a higher resolution.

2. If you have, insert 1-1 photos from a typical mobile and a fixed dune.

3. You need to report in the methods, that simple linear models (simple linear regression) were used for statistical hypothesis testing in R. Furthermore, creating statistical inference from the averaged the 7 time points is not really reasonable. Mixed models or generalized linear mixed models should have been used with temporal autocorrelation in this case!
Additionally, evaluating species richness, and diversity indices with simple linear models suggests some inconsistency, since the distribution of these response variables dont fulfill the assumptions of normality (species richness are count data, simpson index can be "truncated" into the 0-1 interval that can be modeled with Beta regression).

4. Figure 2: Use the same scale and limits on the Y axis for figure 2a) and 2b)

5. Figure 3: Use the same scale and limits on the Y axis for each of the figure pairs.

6. Figure 3b): The error ribbon is missing from the regression lines of the open areas in the case of Shannon and Simpson indices.

7. Figure 3 caption: "Based on the significance of ...."
Extract the last 3 lines from the caption and put it in the results, since these are important information and belong to the results!

8. Figure 4: Recommend to split figure 4 and create two new plots from fig4 a+b and fig4 c+d+e. The current strucure is just too small and hard to read.

9. Figure 5: What happened with the error ribbons in the first 4 plots in row a) and b) ?

10. Figure 5: Use the same scale and limits on the X axis for each of the figure pairs.

line 40: "Both abiotic (e.g.spatial heterogeneity)..."  
Spatial heterogeneity of what? Readers can not decide whether it is abiotic or biotic if it was not specified in detail.

line 74: some information about the elevation in general would be useful

line 99: "...Nizzaim due system..."
should be "Nizzaim dune system"

line 107-114: Considering those two unsuccessful techniques, does the removal of A.monosp. provides a real solution to this problem? I mean, what would be the most efficient method to conserve mobile dunes or even reverse the process in semi-fixed ones? Do we need to intervene at all? I missed your ideas about this in the conclusions.

line 135: "Quadrats were placed randomly..."
How did you designate these random locations? Randomly choosing the location of the quadrats on a map before the studies (e.g. with a software), or in the field during fieldwork?

line 141-146: This text should be presented in the results chapter unless if these were the criteria to classify dune types. If these are not from your results, please provide a citation.

line 142: Ammophila arenaria should be marram grass or European beachgrass instead of "maritime grass"

line 145: "...dunes have 31-50% plant cover..." .
You mean perennial plant cover?

line 149: Recommend to create a list about the prevalent species and add it as an appendix to the manuscript! (english name + scientific name + maybe average cover).

line 150: How did you get to this number of sites? (n=117)?

line 157-157:"...of total plant dunes and patches cover of the community over different years."
-> total plant cover of the community in dunes and patches over different years.  sounds more clear

line 159-161: Explain this index more in detail ("log var ratio formula"), what does these results tell us when the ratio is high and what if the ratio is low? Maybe an equation is also suitable here just to get more insight!

line 167:  + ggplot2 was used for data visualization right?

line 178: "The differences in species number among dunes of the same type are evident in the semi-fixed dunes."
I dont think so. Since these are averaged data and we cannot see its deviations from the mean we cannot say that they were different. Maybe only the lowest one can be different from the other six and considered as an outlier, but no further information was reported about them in the manuscript! Especially if nothing was reported about the within-group variance. (That is why I recommended to use mixed models for this.)

line 179: "Similar trends were observed..."
Trend is not a good word here in this context rather relationship or tendency is more relevant

line 180: "...the trends differ..."
Trend is not a good word here in this context rather relationship or tendency is more relevant

line 181: the word "both" doesn't needed (redundant)

line 182: "parallel to"
with or parallel with

line 183: "No significant differences were found between both."
Report the result (coefficients, p value etc), from the linear model.
I see that the two lines and the default error ribbons are overlapping, but model results needs to be reported too!

line 197-198: "The temporal stability of annual plant assemblages at the dune scale was not affected by the perennial plants’ coverage..."
It sounds a bit strange that stability wasnt affected by plant cover, since plant cover was the main component when stability was calculated right?

line 201: "...open patches and beneath-the-shrubs patches..."
This clause is not needed.

line 236-237: "...and that the shrubs do not facilitate annual plant coverage beneath them during dune fixation process..."
Sounds logic, but we heard a lot about the facilitating and ecosystem engineer functions of woody plants in the introduction, so these results sound a bit contradictory, this needs to be clarified at the end!
Do woody plants facilitate or hinder the colonization of annual plants then?

Reviewer 3 Report

This work shows that annual plant cover, species richness, and diversity increased as the dunes became more stable. The patches under the shrubs had the most effect on these changes. In the same way, different types of dunes and patches had different plant communities. 

Overall the manuscript is well and can be accepted after a careful English language check.